# Associations of Chronic Diabetes Complications and Cardiovascular Risk with the Risk of Obstructive Sleep Apnea in Patients with Type 2 Diabetes

**DOI:** 10.3390/jcm11154403

**Published:** 2022-07-28

**Authors:** Diana Cristina Protasiewicz Timofticiuc, Ionela Mihaela Vladu, Adela-Gabriela Ștefan, Diana Clenciu, Adina Mitrea, Vlad Pădureanu, Ion Cristian Efrem, Ileana-Diana Diaconu, Adina Turcu, Tiberiu Ștefăniță Țenea-Cojan, Anca Mihaela Hâncu, Maria Forțofoiu, Oana Mirea Munteanu, Maria Moța

**Affiliations:** 1Doctoral School, University of Medicine and Pharmacy of Craiova, 200349 Craiova, Romania; diana_protasiewicz@yahoo.com (D.C.P.T.); dianadiaconu655@yahoo.com (I.-D.D.); mmota53@yahoo.com (M.M.); 2Department of Diabetes, Nutrition and Metabolic Diseases, County Clinical Emergency Hospital of Craiova, 200642 Craiova, Romania; 3Department of Diabetes, Nutrition and Metabolic Diseases, Faculty of Medicine, University of Medicine and Pharmacy of Craiova, 200349 Craiova, Romania; dianaclenciu@yahoo.com; 4Calafat Municipal Hospital, 205200 Calafat, Romania; adela.firanescu@yahoo.com; 5Department of Diabetes, Nutrition and Metabolic Diseases, Clinical Municipal Hospital “Philanthropy” of Craiova, 200143 Craiova, Romania; 6Department of Medical Semiology, Faculty of Medicine, University of Medicine and Pharmacy of Craiova, 200349 Craiova, Romania; vlad.padureanu@umfcv.ro; 7Department of Internal Medicine, County Clinical Emergency Hospital of Craiova, 200642 Craiova, Romania; 8Department of Internal Medicine and Medical Semiology, Faculty of Dentistry, University of Medicine and Pharmacy of Craiova, 200349 Craiova, Romania; cristian.efrem@umfcv.ro; 9Department of Internal Medicine, Clinical Municipal Hospital “Philanthropy” of Craiova, 200143 Craiova, Romania; 10Department of Pediatric Pneumology, “Marius Nasta” National Institute of Pneumophtisiology, 050159 Bucharest, Romania; 11Faculty of Dentistry, University of Medicine and Pharmacy of Craiova, 200349 Craiova, Romania; adinaturcu14@yahoo.com; 12Infectious Diseases Hospital “Victor Babes”, 200515 Craiova, Romania; 13Department of General Surgery, Faculty of Medicine, University of Medicine and Pharmacy of Craiova, 200349 Craiova, Romania; doctortts@yahoo.com; 14Department of General Surgery, C.F. Clinical Hospital, 200374 Craiova, Romania; 15Nutriscience Clinic, 011761 Bucharest, Romania; ancahancu@gmail.com; 16Department of Emergency Medicine, Faculty of Medicine, University of Medicine and Pharmacy of Craiova, 200349 Craiova, Romania; 17Department of Emergency Medicine, Clinical Municipal Hospital “Philanthropy” of Craiova, 200143 Craiova, Romania; 18Department of Cardiology, University of Medicine and Pharmacy of Craiova, 200349 Craiova, Romania; oana.munteanu@umfcv.ro; 19Department of Cardiology, County Clinical Emergency Hospital of Craiova, 200642 Craiova, Romania

**Keywords:** obstructive sleep apnea, STOP-Bang score, type 2 diabetes, cardiovascular risk

## Abstract

Background: Type 2 diabetes mellitus (T2DM) is associated with increased mortality and morbidity, including cardiovascular diseases and obstructive sleep apnea (OSA). The aim of this study was to assess the associations between cardiovascular risk, chronic diabetes complications and the risk of OSA in adult patients with T2DM. Methods: The study included 529 patients with T2DM in whom moderate-to-severe OSA risk was assessed using the STOP-Bang questionnaire, dividing the subjects into two groups: group 1: STOP-Bang score <5, and group 2: STOP-Bang score ≥5, respectively. In all the subjects, cardiovascular risk was assessed using the UKPDS risk engine. Statistical analysis was performed using SPSS 26.0, the results being statistically significant if *p* value was <0.05. Results: 59% of the subjects scored ≥5 on the STOP-Bang questionnaire. We recorded statistically significant differences between the two groups regarding diabetes duration, HbA1c, HOMA-IR, albuminuria, as well as cardiovascular risk at 10 years for both coronary heart disease (CHD) and stroke (*p* < 0.05). Furthermore, through logistic regression, adjusting for confounding factors, we demonstrated that the STOP-Bang score ≥ 5 is a risk factor for 10-year fatal and nonfatal CHD risk. Conclusions: It is extremely important to screen and diagnose OSA in patients with T2DM, in order to improve the primary and secondary prevention of cardiovascular events in these patients.

## 1. Introduction

Obstructive sleep apnea (OSA) is an intermittent, partial or complete obstruction of the upper airways that occurs during sleep, leading to intrathoracic pressure changes, intermittent hypoxemia as well as fragmented sleep, which impair patients’ quality of life, with significant impacts on both physical and mental health [1]. What is more, this sleep disorder associates functional and structural metabolic disorders. As a result, this condition is a major public health problem [2,3].

Although OSA is a very common condition, it remains largely undiagnosed [1,4]. The reported prevalence of OSA is increasing, partially because of the increase in obesity, which is an important risk factor for OSA, as well as due to the improved sensitivity of the methods used for the screening and diagnosis of OSA [1]. Nocturnal polysomnography, which diagnoses OSA based on the apnea-hypopnea index (AHI), is the gold standard for diagnosing this disorder. Senaratna et al. reported in their review that when the cut-off used for the diagnosis of OSA is an AHI of ≥5 events/hour, OSA has a prevalence of 9–38% in the general population, while an AHI cut-off of ≥15 events/hour is associated with an OSA prevalence of 6–17% in the general population [5]. The prevalence of OSA in certain groups of patients that are at high risk of developing OSA is higher. The risk factors for OSA include: retrognathia, family history of OSA, cardiovascular diseases (treatment-resistant hypertension, atrial fibrillation, congestive heart failure, stroke), acromegaly, Down syndrome, polycystic ovary syndrome, obesity (especially stage 2 and 3 obesity) and type 2 diabetes mellitus (T2DM) [6]. T2DM and OSA are two conditions that increase their mutual risk [2,7]. It was estimated that in patients with T2DM, OSA has a prevalence of 50–80% [8,9].

Although polysomnography is the most accurate method used for the diagnosis of OSA, this investigation cannot be used in all patients that are at high risk for sleep apnea, as it is expensive, time consuming and inconvenient for the patients. In this light, especially in settings with low resources that have a limited capability for referring at-risk patients for polysomnography, screening questionnaires become very useful in daily clinical practice [1]. The most used questionnaires are Epworth sleepiness scale [10], Berlin [11] and STOP-Bang questionnaires [12]. A meta-analysis from 2017 [13], analyzing the performance of the three scores for the detection of OSA, reported that STOP-Bang score had the highest sensitivity for the detection of moderate (90%) and severe (93%) OSA.

The STOP-Bang questionnaire was validated in the general population as well as for patients with obesity [14]. Two studies conducted in Asian [15] and Caucasian [16] patients with T2DM also proved the utility of the STOP-Bang score in this group of patients. Moreover, a meta-analysis including data from 1894 patients demonstrated that the STOP-Bang questionnaire is an effective tool for OSA screening in patients with cardiovascular risk factors [17]. The sensitivity of the STOP-Bang questionnaire is 90% for the identification of OSA risk when using a cut-off of ≥3, while the use of a cut-off of ≥5, increases the sensitivity to identify the subjects at risk of moderate-to-severe OSA, but at the cost of a decreased specificity [18]. Furthermore, in a recent prospective study [18] it was concluded that STOP-Bang scores ≥5 were associated with high cardiovascular mortality within three years follow-up.

OSA involves increased mortality and morbidity, being frequently associated with hypertension (HTA), heart failure (HF), coronary heart disease (CHD), arrhythmias, fatal myocardial infarction (MI) and stroke [19,20,21,22]. Patients with T2DM also have an increased cardiovascular risk and although there were important therapeutic advances which have improved the outcomes of these patients, cardiovascular diseases are the main causes of mortality in patients with T2DM [23].

It is important to assess the cardiovascular risk in both patients with T2DM and patients with OSA, in order to prevent adverse cardiovascular outcomes in these subjects. Regarding the tools that at the moment are used for the evaluation of cardiovascular risk, there are various algorithms that can be used [24], though not all have been validated in patients with T2DM. One of the most widely used models for cardiovascular risk assessment in patients with T2DM remains the United Kingdom Prospective Diabetes Study (UKPDS) risk engine [25,26].

Taking into account the high cardiovascular risk of patients diagnosed with T2DM, as well as the necessity to actively screen for OSA in these patients, the aim of our study was to assess the associations between cardiovascular risk and chronic diabetes complications in patients with T2DM and the risk of moderate-to-severe OSA, assessed by using the STOP-Bang questionnaire.

## 2. Materials and Methods

### 2.1. Participants

We conducted an epidemiological, non-interventional, cross-sectional study, over 3 years, between 2018 and 2021. The study sample size was calculated using calculator.net [27] for cross sectional studies. We used a confidence level of 95% with a corresponding z-score of 1.96, selecting a margin of error of 5%. A frequency of moderate-to-severe OSA risk in the study population of 59% was estimated together with a study power of 80%. We also assumed that about 30% of the subjects could decline participation in the study, as well as 12% of the subjects with incomplete STOP-Bang questionnaires. Taking these data into account, we estimated that at least 529 patients were required in order to have a confidence level of 95% [8,9]. As presented in Figure 1, 553 answered the STOP-Bang questionnaire, 529 having complete data were enrolled in the study. According to the results obtained from the STOP-Bang questionnaire, the study subjects were divided into 2 groups, as follows:-Group 1—217 patients with STOP-Bang score <5;-Group 2—312 patients with STOP-Bang score ≥5 (patients with moderate-to-severe OSA risk [18]).

The study inclusion criteria were: patients with T2DM aged over 18 years recruited from the Outpatient Departments of Diabetes, Nutrition and Metabolic Diseases of the Clinical County Emergency Hospital of Craiova and the Municipal Hospital “Philanthropy” of Craiova.

The following exclusion criteria were used in this study: patients aged under 18 years, type 1 diabetes or other forms of diabetes, the presence of severe comorbidities, such as other respiratory diseases, neoplasms, central sleep apnea, autoimmune disorders or disorders that can alter the activity of the hypothalamic-pituitary-adrenal axis, psychiatric pathology, substance abuse at present or in the past, patients who have undergone bariatric surgery, as well as other conditions that could influence the results of the investigations carried out.

All the participants were voluntarily included in the study, after signing the informed consent. The present study was conducted in accordance with the ethical principles set out in the Helsinki Declaration—updated, as well as in accordance with Good Clinical Practice (GCP) and respected the right to confidentiality and integrity. The subjects had the option to withdraw from this study at any time. The study was performed with the approval of the Ethics Committee of the County Clinical Emergency Hospital of Craiova, Romania (12297/16 March 2022) and by the Ethics Committee of the University of Medicine and Pharmacy of Craiova, Romania (43/24 March 2022).

Anamnestic data as well as significant medical data obtained during the patients’ physical examination were recorded. Demographic and anthropometric data were also recorded. Weight and height were measured and used in order to calculate the body mass index (BMI) according to the formula: BMI = weight (kilograms)/height^2^ (meters). The patients’ nutritional status was evaluated according to BMI, using the World Health Organization criteria [28]. The following circumferences were measured: waist circumference (WC) at the midpoint between the upper iliac crest and the lower border of the rib cage; the hip circumference (HC) over the femoral trochanters and the neck circumference (NC) below the cricoid cartilage. Abdominal obesity was also assessed using the waist to hip ratio (WHR) calculated using the formula WC (cm)/HC (cm). The waist to height ratio (WHtR), calculated using the formula WC (cm)/height (cm), was another parameter that was used for the assessment of visceral adiposity as well as cardiovascular and metabolic risk. The NC height ratio (NHR) was also calculated using the formula NC (cm)/height (cm).

Blood pressure (BP) was measured using an automatic sphygmomanometer for sitting patients after 10 min rest. Hypertension was diagnosed if the subjects either had systolic BP ≥ 140 mmHg, diastolic BP ≥ 90 mmHg, antihypertensive drug treatment or in combination, in patients with personal history of hypertension.

Cardiovascular disease (CVD) was defined as ischemic heart disease (IHD), stroke/transient ischemic attack (TIA), peripheral vascular disease (PVD) or atrial fibrillation (AFi). Atherosclerotic cardiovascular disease (ASCVD), macrovascular chronic diabetes complication, was defined by the presence of IHD, stroke/AIT or PVD. All subjects with ejection fraction (EF) <50% were diagnosed with heart failure (HF).

Microvascular chronic diabetes complications were also assessed. The diagnosis of diabetic retinopathy was established after dilated fundus examination [29]. Diabetic peripheral neuropathy was assessed based on the presence of characteristic symptoms (pain, dysesthesias, numbness) and a combination of temperature sensation (for small fiber function) and vibration sensation (for large-fiber function) tests, as it is recommended by the American Diabetes Association [29]. The presence of chronic kidney disease was evaluated using Kidney Disease: Improving Global Outcomes (KDIGO) recommendations [30].

### 2.2. Laboratory Exams

Venous blood samples were collected from all participants, following the standard procedure. High-performance liquid chromatography assay (HPLC) was the method used for glycated hemoglobin (HbA1c) assessment. Fasting plasma glucose (FPG) and fasting insulinemia were also collected and the Homeostatic Model Assessment for Insulin Resistance (HOMA-IR) was calculated using the formula: (FPG (mg/dL) × fasting insulinemia (mU/L))/405. Serum creatinine was measured and the estimated glomerular filtration rate (eGFR) was calculated according to the CKD-EPI formula [31]. Albuminuria was also determined.

### 2.3. Evaluation of Obstructive Sleep Apnea Syndrome risk

The OSA risk was evaluated using the STOP-Bang questionnaire [32]. This score includes eight dichotomous “yes” or “no” questions, referring to the clinical features of sleep apnea (snoring, observed apnea, fatigue, hypertension, age, NC, BMI and male gender). The answer “yes” to each question was scored 1. The moderate-to-severe OSA risk was defined if STOP-Bang score was ≥5 [18].

### 2.4. Evaluation of Cardiovascular Risk

The 10-year cardiovascular risk of fatal and nonfatal CHD, respectively, fatal and nonfatal stroke, was calculated using the UKPDS risk engine program, originally developed by Oxford University. The program calculates the individual 10-year risk of fatal and nonfatal stroke and CHD taking into account the duration of T2DM, patient’s age, gender, ethnicity, smoking status, presence or absence of FiA, as well as the value of systolic blood pressure, HbA1c, total cholesterol and high-density lipoprotein cholesterol (HDLc) [21,33]. The risk was calculated only for patients that did not have a personal history of myocardial infarction or stroke, as the UKPDS risk engine is recommended for the risk assessment of future coronary events and stroke in patients with T2DM without previous acute events.

### 2.5. Statistical Analysis

The recorded parameters were registered in an Excel database, then transferred to Statistical Package for the Social Sciences (SPSS) version 26.0 (SPSS Inc., Chicago, IL, USA), encoded and then analyzed based on the value of the STOP-Bang score. The Kolmogorov–Smirnov test was used to assess normality of continuous variables, data with normal distribution being presented as mean ± standard deviation (SD), while data with abnormal distribution were presented as median and interquartile range (IQR).

The statistical significance of the differences between groups was evaluated using the Mann–Whitney U test to compare the medians. Categorical variables were compared using the chi-square test. The statistical analysis of the data was completed using logistic regression. The results obtained from the statistical analysis performed were considered significant if a value of *p* < 0.05 was obtained.

## 3. Results

In our study, 59% of the patients with T2DM (312 patients) presented a STOP-Bang score ≥5, which puts them at risk of moderate-to-severe OSA. We assessed the 10-year fatal and nonfatal CHD and stroke risk using the UKPDS risk engine comparatively between the two study groups. This analysis proved that patients with STOP-Bang scores above 5 are at a significantly higher risk of developing fatal and non-fatal CHD (Table 1 and Figure 2). 

After adjusting for age, gender, smoking status, BMI, WC, NC, T2DM duration, HbA1c and established ASCVD (with the exception of personal history of myocardial infarction and stroke, patients with previous events being excluded from this analysis), through logistic regression we established that STOP-Bang score ≥5 was associated with moderate-increased 10-year risk of nonfatal CHD and fatal CHD but not with the 10-year risk of nonfatal and fatal stroke (Table 2).

The statistical analysis regarding parameters associated with T2DM and diabetes chronic complications are presented in Table 3. A statistically significant higher percentage of patients with poor control of T2DM (HbA1c ≥ 7%) was found in the group of patients with STOP-Bang score ≥5 (*p* = 0.002) (Figure 3a). Furthermore, we also observed a statistically higher percentage of patients with both microalbuminuria and macroalbuminuria within the subjects with STOP-Bang scores ≥5 (*p* = 0.005) (Figure 3b).

We used logistic regression in order to evidence the factors associated with moderate-to severe OSA risk in patients with T2DM (BMI, WC, male gender, chronic diabetes complication, T2DM duration, HbA1c, HOMA-IR, smoking status), proving that WC, male gender and the presence of established ASCVD are risk factors for moderate-to-severe OSA risk (Table 4).

The relationships between smoking status, STOP-Bang score ≥5, established ASCVD and 10-year risk of CHD were also analyzed. Of the subjects enrolled in the study, 13.8% (73) were smokers, while a percentage of 18.7% (99) were former smokers. Interestingly, we found no statistically significant differences regarding the frequency of active and former smoking between the two study groups. Moreover, we observed the same pattern for patients with T2DM and established ASCVD. Only the analysis regarding smoking status and 10-year fatal and nonfatal CHD risk showed statistical significance (*p* = 0.023, respectively, *p* = 0.001), as shown in Figure 4a,b.

## 4. Discussion

Recently, particular attention has been paid in the literature to the association between OSA, diabetes mellitus and cardiovascular risk, but very few authors have assessed cardiovascular risk using the UKPDS risk engine in patients with OSA. In our study population, using STOP-Bang score ≥5 as a screening method for moderate-to severe OSA, we observed a frequency of the OSA risk of 59%, similar to the prevalence of OSA in patients with T2DM reported in the literature [8,9]. Numerous authors have used 10-year cardiovascular risk models, such as Framingham risk score (FRS) or Systematic Coronary Risk Evaluation (SCORE), also demonstrating an increased cardiovascular risk parallel to OSA severity [19]. OSA has been associated with increased morbidity and mortality due to cardiovascular disease, especially stroke and CHD [19,34].

We compared 10-year cardiovascular risk in patients with T2DM according to the STOP-Bang score. Similar to our study, other studies have shown that the STOP-Bang score can be used to identify patients at risk of developing fatal or nonfatal CHD as well as fatal or nonfatal stroke [17,18,35]. In a recent prospective study that enrolled 435 patients, it was demonstrated that a STOP-Bang score ≥5 was associated with increased all-cause mortality and cardiovascular mortality over 36 months follow-up (95 patients died, 34 of them due to cardiovascular events), with a hazard ratio of 3.12 [18]. Furthermore, the same authors demonstrated in an observational study from 2019 that altered STOP-Bang score is associated with an increased cardiovascular risk assessed using the American Heart Association cardiovascular risk calculator [36]. The same study also observed major cardiovascular events (MACE) in the study population over a one-year follow-up, demonstrating a statistically significant association between MACE and OSA risk [36]. 

In our study, a STOP-Bang score ≥5 represented a risk factor for 10-year risk of fatal and nonfatal CHD, estimated using the UKPDS risk engine, after adjustment for many possible confounders (age, gender, smoking status, BMI, WC, NC, T2DM duration, HbA1c and established ASCVD). We consider the fact that STOP-Bang score above 5 is associated with increased cardiovascular risk as important, as it is easier to perform this questionnaire even in patients with T2DM, the UKPDS risk engine being more complex and requiring more clinical and paraclinical data that are not always available in primary care, where we recommend OSA screening using the STOP-Bang questionnaire in patients with T2DM. Previous studies that have validated the STOP-Bang questionnaire in patients with T2DM have reported conflicting results regarding the relationship between HbA1c and the OSA diagnosis [15,16]. The authors of one of these studies suggested that it might be useful to include HbA1c in the STOP-Bang questionnaire for OSA screening in patients with diabetes [15]. In our study, the HbA1c levels in subjects with STOP-Bang scores above 5 were statistically significantly higher compared to subjects with STOP-Bang scores below 5, suggesting the necessity to screen for OSA in patients with T2DM and suboptimal HbA1c levels.

Furthermore, studies have demonstrated that there is a correlation between OSA and insulin resistance, recommending screening for insulin resistance in patients with OSA [37,38,39,40]. However, one of these studies concluded that insulin resistance assessed by HOMA-IR, was not associated with AHI but rather with BMI [40]. A recent study which enrolled patients with T2DM demonstrated that higher HOMA-IR values are associated with increased cardiovascular risk [41]. In our study, we also observed that HOMA-IR was statistically significantly higher in patients with T2DM and OSA, diagnosed by STOP-Bang score ≥5, thus confirming the results reported in previous studies [37,38,39]. What is more, the duration of T2DM was also associated with moderate-to-severe OSA risk, leading us to recommend OSA screening in patients with long-standing T2DM. Moreover, our study also demonstrated a statistically significant higher frequency of microvascular chronic diabetes complications (chronic kidney disease and diabetic retinopathy) in patients with OSA risk. Recent studies have proved an association between OSA and diabetic nephropathy and impairment of renal function in patients with T2DM, concluding that the association of OSA with T2DM is an independent risk factor for chronic kidney disease [42,43]. However, as the mechanisms associating OSA and T2DM are still not completely elucidated, further studies, such as the ongoing trial conducted by Antza et al. [44], are needed in order to reduce the progression of both pathologies in patients with T2DM.

Regarding macrovascular diabetes complications, in our study, 61.4% of the subjects enrolled in the study presented established ASCVD, with a statistically significant higher frequency in patients with STOP-Bang scores ≥5 (67% vs. 53.5%), confirming data reported in the medical literature [45,46]. Additionally, in a study from 2021, which enrolled 1204 patients with peripheral arterial disease, it was concluded that the diagnosis of OSA is independently associated with worse health status over time [46].

In a study from 2008, Chung et al. [47] proved that STOP-Bang score has a sensitivity of 100% for AHI ≥ 30; however, the small specificity of 37.0% means that there are also many false positive cases reported, suggesting that even though the STOP-Bang questionnaire is a very quick and easy tool to apply for the screening of OSA, in order to properly diagnose and treat this category of patients they should be referred for polysomnography or home sleep apnea testing (HSAT), as is recommended by the American Academy of Sleep Medicine (AASM) [48]. However, many patients unfortunately refuse undergoing such expensive investigations, and in these conditions, undiagnosed cases can significantly increase the burden on the public health system and also decrease the quality of life of patients, thus necessitating the acceptance of a screening method with high sensitivity, even at the cost of low specificity, such as the STOP-Bang questionnaire [17].

Over time, numerous studies have established a close association between OSA and cardiovascular diseases (stroke, hypertension, HF, FiA, cerebrovascular disease, CHD) [17,49,50]. Epidemiological studies have shown that obesity increases the risk of developing both cardiovascular and metabolic disorders. OSA thus becomes an independently modifiable risk factor for cardiovascular disease, and may induce its development or worsening [5,50,51].

Furthermore, taking into account the fact that studies assessing the links between OSA and smoking status are controversial [52,53,54], we also tried to identify whether, in our study population, there was a relationship between smoking and OSA risk. An important percentage of patients with T2DM included in the study were active (13.8%) or former (18.7%) smokers, comparable to the high prevalence of active smokers (18%) and former smokers (30.8%) estimated in the adult Romanian population by the PREDATORR epidemiological study [55]. With regard to the relationships between smoking status and both fatal and nonfatal CHD 10-year risk in patients with T2DM, we found statistically significant associations, as was expected and reported by other studies [56,57]. Although, in this study, we could not prove that active and former smoker status were associated with OSA risk, smoking remains a modifiable risk factor for pulmonary and metabolic diseases and patients should be encouraged to cease this habit.

The results of our study confirm previous findings regarding the relationships between T2DM and OSA. It is recognized that this is a bidirectional relationship, with patients diagnosed with OSA having a higher probability of developing T2DM, while more than half of patients with T2DM also being diagnosed with OSA [58]. 

Given the importance of OSA on cardiovascular risk and the already increased cardiovascular risk of patients with T2DM, this category of patients should be screened with the STOP-Bang questionnaire and the subjects with abnormal scores (≥3) should be further referred for polysomnography or HSAT. Strengths of our study include the large sample size as well as the assessment of cardiovascular risk by using the UKPDS risk engine, a tool with recognized utility in patients with T2DM. The limitations of the study include the fact that this study was conducted in a limited geographical area and cannot be extrapolated to the general population. Due to the random inclusion of the subjects in this study (the patients were consecutively enrolled in the study), the percentage of subjects ≥60 years of age was 57.08%; this could be a limitation given the increasingly early diagnosis of T2DM with the intensification of prevention and early diagnosis programs for this condition. It is well known that patients with T2DM have a risk of early cardiovascular damage, which may occur approximately 14.6 years earlier compared to those without T2DM [59]. However, the results obtained can be applied to populations with similar characteristics to the participants in the present study.

Despite these limitations, the results of our study suggest that sleep disorders could be considered an important modifiable risk factor in patients with T2DM and cardiovascular disorders.

## 5. Conclusions

OSA is frequently underdiagnosed, with polysomnography being a rather expensive investigation. Furthermore, the STOP-Bang score seems to have a great utility for both OSA screening as well as a cardiovascular assessment, higher values being associated with established cardiovascular diseases as well as higher 10-year cardiovascular risk. Patients with T2DM should be screened early in the evolution of the disease for OSA, and in the cases of high STOP-Bang scores they should be referred to a pulmonologist for the confirmation of this diagnosis through polysomnography or HSAT. Moreover, as patients with T2DM are recognized for their increased risk of developing both OSA and cardiovascular disorders and, in this light, in order to improve cardiovascular outcomes in these patients, it is imperative to intervene in all cardiovascular risk factors as soon as they are detected. It is also extremely important that all physicians, regardless of specialty, know the association between the two conditions, in order to improve both primary and secondary prevention of cardiovascular events and to provide patients with T2DM with adequate access to medical education and specialized treatment.

## Figures and Tables

**Figure 1 jcm-11-04403-f001:**
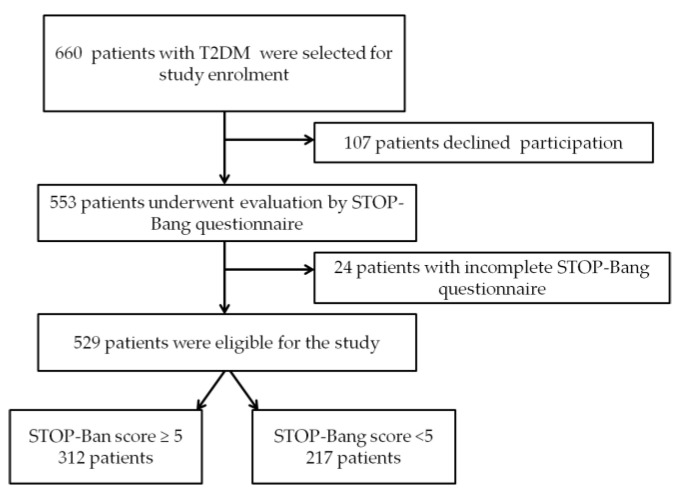
Study flow chart. T2DM: type 2 diabetes mellitus; OSA: obstructive sleep apnea.

**Figure 2 jcm-11-04403-f002:**
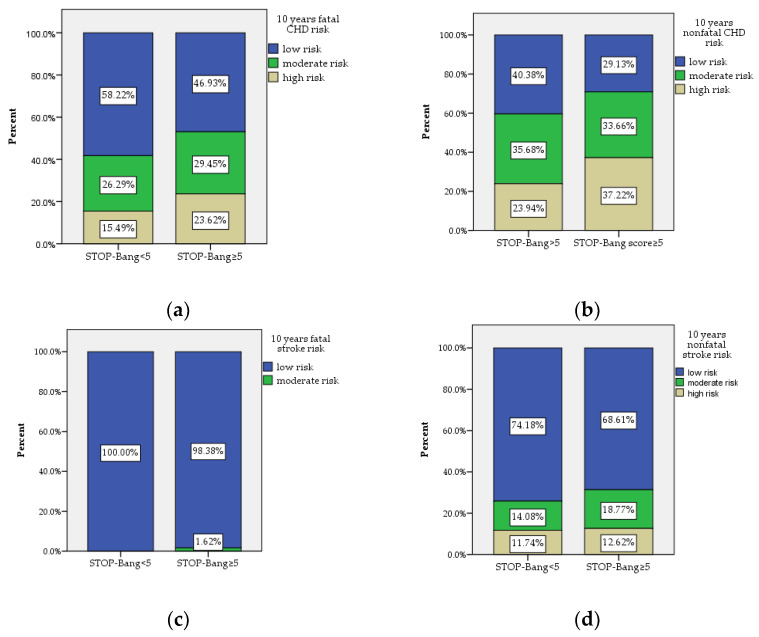
The frequency of 10-year cardiovascular risk calculated using the UKPDS risk engine evidencing that (**a**) patients with T2DM and STOP-Bang scores ≥5 are at statistically significant higher risk of developing fatal CHD (*p* = 0.022); (**b**) patients with T2DM and STOP-Bang scores ≥5 are at statistically significant higher risk of developing nonfatal CHD (*p* = 0.003); (**c**) there was no statistically significant difference between the two study groups regarding fatal stroke; (**d**) no statistically significant differences were found between the two study groups.

**Figure 3 jcm-11-04403-f003:**
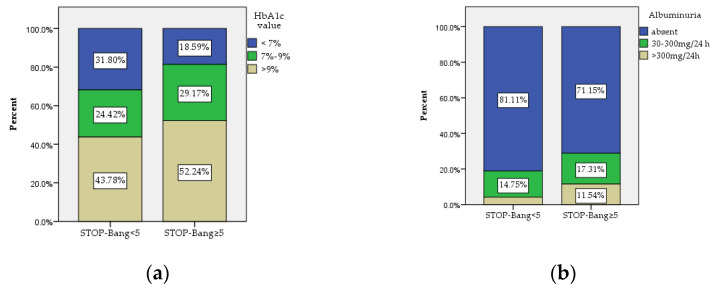
The distribution within the two study groups of (**a**) HbA1c: patients with STOP-Bang scores ≥5 had in a significantly higher percentage HbA1c > 7% (*p* = 0.002); (**b**) albuminuria: patients with STOP-Bang scores ≥5 presented in a significantly higher percentage albuminuria (*p* = 0.005). HbA1c: glycated hemoglobin.

**Figure 4 jcm-11-04403-f004:**
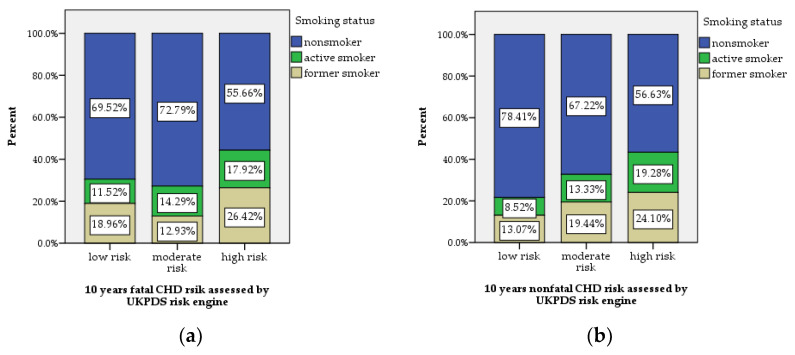
The distribution of smoking status in the study population (**a**) patients with T2DM and high risk of fatal CHD were in a statistically significant higher percentage for active and former smokers (*p* = 0.023) and (**b**) active and former smokers were statistically more frequent within the patients with T2DM and moderate and high risk of nonfatal CHD (*p* = 0.001).

**Table 1 jcm-11-04403-t001:** The 10-year cardiovascular risk assessed by the UKPDS risk engine stratified by STOP-Bang score.

Characteristic	Total	STOP-Bang < 5	STOP-Bang ≥ 5	*p*
Fatal CHD 10-year risk (median (IQR)) *	14.00 (19.0)	12.20 (16.7)	15.50 (20.9)	0.001
Nonfatal CHD 10-year risk (median (IQR)) *	20.70 (22.6)	19.00 (18.7)	22.70 (24.8)	0.001
Fatal stroke 10-year risk (median (IQR)) *	1.30 (2.2)	1.20 (2.0)	1.30 (2.2)	0.187
Nonfatal stroke 10-year risk (median (IQR)) *	8.00 (13.0)	7.70 (12.7)	8.40 (13.2)	0.261

* Variables with abnormal distribution, presented as median (IQR). CHD: coronary heart disease; IQR: interquartile range; UKPDS risk engine: United Kingdom Prospective Diabetes Study (UKPDS) risk engine.

**Table 2 jcm-11-04403-t002:** Logistic regression for the association of STOP-Bang score ≥5 with 10-year cardiovascular risk assessed by the UKPDS risk engine.

10-Year Risk	OR † [95% CI]	*p* †	OR * [95% CI]	*p* *
Moderate-increased fatal CHD 10 years risk	1.275 [1.129; 1.440]	<0.001	1.576 [1.108; 2.241]	0.011
Moderate-increased nonfatal CHD 10 years risk	1.311 [1.151; 1.493]	<0.001	1.648 [1.141; 2.380]	0.008
Moderate-increased fatal stroke 10 years risk	1.891 [0.987; 3.622]	0.055	0.734 [0.332; 1.622]	0.443
Moderate-increased nonfatal stroke 10 years risk	1.141 [1.002; 1.300]	0.046	1.531 [0.843; 2.781]	0.161

† Unadjusted regression; * Adjustment for age, gender, smoking status, BMI, WC, NC, T2DM duration, HbA1c and established ASCVD. ASCVD: atherosclerotic cardiovascular disease; BMI: body mass index; CHD: coronary heart disease; CI: confidence interval; NC: neck circumference; OR: odds ratio; T2DM: type 2 diabetes mellitus; UKPDS risk engine: United Kingdom Prospective Diabetes Study risk engine.

**Table 3 jcm-11-04403-t003:** Comparative analysis of the characteristics associated with T2DM.

Characteristic	Total	STOP-Bang < 5	STOP-Bang ≥ 5	*p*
T2DM duration, years (median (IQR)) *	7.00 (11)	6.00 (10)	8.00 (12)	<0.001
HbA1c, % (median (IQR)) *	8.80 (3.60)	8.30 (3.59)	9.20 (3.42)	<0.001
HOMA-IR (median (IQR)) *	2.08 (0.52)	1.90 (0.34)	2.31 (1.06)	<0.001
Chronic kidney disease, no (%)	168 (31.8%)	56 (10.6%)	112 (21.2%)	0.014
Diabetic retinopathy, no (%)	202 (38.2%)	65 (30%)	137 (43.9%)	0.005
Diabetic peripheral neuropathy, no (%)	448 (84.7%)	184 (84.8%)	264 (84.6%)	0.956
ASCVD, no (%)	325 (61.4%)	116 (53.5%)	209 (67.0%)	0.002

* Variables with abnormal distribution, presented as median (IQR). ASCVD: atherosclerotic cardiovascular disease; HbA1c: glycated hemoglobin; HOMA-IR: Homeostatic Model Assessment for Insulin Resistance; IQR: interquartile range; T2DM: type 2 diabetes mellitus.

**Table 4 jcm-11-04403-t004:** Predictors for moderate-to-severe OSA risk in patients with T2DM.

Analyzed Parameter	OR [95% CI]	*p*
Waist circumference	0.871 [0.824; 0.920]	<0.001
Male gender	3.124 [1.183; 8.250]	0.021
Established ASCVD	0.191 [0.068; 0.534]	0.002

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
