# Peer review of "Associations of Chronic Diabetes Complications and Cardiovascular Risk with the Risk of Obstructive Sleep Apnea in Patients with Type 2 Diabetes"

_jcm, 2022, doi:10.3390/jcm11154403_

Round 1

Reviewer 1 Report

- in the introduction are not mentioned the different treatment options and how they reduce the comorbidities related to OSA. Please discuss and cite doi:10.1111/resp.13144. and doi:10.1016/j.amjoto.2021.103197.

- adopt the strobe guidelines to improve the quality of the structure

- OSA patients are more likely than non-OSA populations to develop type 2 diabetes, while more than half of type 2 diabetes patients suffer from OSA. Similar to Western countries, in the East Asian population, the association between these two disorders has also been reported. CPAP is the primary treatment for OSA, but the effect of CPAP on comorbid diabetes has not been established. CPAP improved glucose metabolism determined by the oral glucose tolerance test in OSA patients, and several studies have shown that CPAP improves insulin resistance, particularly in obese populations undergoing long-term CPAP. Diabetes is associated with other sleep-related manifestations as well, such as snoring and excessive daytime sleepiness. Snoring is associated with the development of diabetes, and excessive daytime sleepiness appears to modify insulin resistance., please discuss and cite doi:10.1111/jdi.12823.

Author Response

Point 1: in the introduction are not mentioned the different treatment options and how they reduce the comorbidities related to OSA. Please discuss and cite doi:10.1111/resp.13144. and doi:10.1016/j.amjoto.2021.103197.

Response 1: We thank the reviewer for pointing this out. We have discussed the treatment options and included the suggested papers in the reference list.

Point 2: adopt the strobe guidelines to improve the quality of the structure.

Response 2: Thank you for this suggestion. We have revisited the structure of the manuscript accordingly, adding the study flow chart, improving the title and material and methods section.

Point 3: OSA patients are more likely than non-OSA populations to develop type 2 diabetes, while more than half of type 2 diabetes patients suffer from OSA. Similar to Western countries, in the East Asian population, the association between these two disorders has also been reported. CPAP is the primary treatment for OSA, but the effect of CPAP on comorbid diabetes has not been established. CPAP improved glucose metabolism determined by the oral glucose tolerance test in OSA patients, and several studies have shown that CPAP improves insulin resistance, particularly in obese populations undergoing long-term CPAP. Diabetes is associated with other sleep-related manifestations as well, such as snoring and excessive daytime sleepiness. Snoring is associated with the development of diabetes, and excessive daytime sleepiness appears to modify insulin resistance., please discuss and cite doi:10.1111/jdi.12823.

Response 3: We absolutely agree, therefore we have modified the Discussions section of the manuscript, including the suggested reference.

Reviewer 2 Report

1.    It says "the aim of our study was to assess cardiovascular risk in patients with T2DM and OSA, assessed by the STOP-Bang questionnaire", however, the study may be assessing the cardiovascular risk of patients with T2DM over the age of 18 who are suspected of having OSA by STOP-Bang questionnaire.

2.    The legend in Figure 1 (a) was not found. There is no description of "significant difference" in Figures 1, 4 and 5. Please correct it so that we can understand it only with the figure without reading the text.

3.    Where can I find the contents of Lines 269-273 in Table1?

4.    The STOP-Bang questionnaire is useful for OSA screening. However, based on "Anesthesiology. 2008 May; 108 (5): 812-21. Doi: 10.1097 / ALN.0b013e31816d83e4." by Chung F et al., the sensitivity of STOP-Bang is 100% for AHI 30, but specificity is 37.0%. This means that there are many false positives, suggesting that many patients have STOP-Bang 3 or higher, even if they are not AHI 30.

5.    As a chronic complication of the target patients, there is already a significant difference in ASCVD between the group with STOP-Bang 5 and the group with STOP-Bang <5. Therefore, it is not surprising that the risk of predicted CHD after 10 years is higher in the group with STOP-Bang 5 than in the group with STOP-Bang <5.

1.    In patients with suspected sleep apnea syndrome, the questionnaire alone has limited diagnostic performance, such as the majority of patients without obstructive sleep apnea being misdiagnosed as obstructive sleep apnea.

2.    For this reason, AASM is highly recommended that do not use questionnaires without polysomnography or HSAT, and do polysomnography or HSAT for adults suspected of having moderate or severe obstructive sleep apnea due to symptoms, etc.

3.    Therefore, even if the cost can be saved, there is a problem in diagnosing OSA only with the STOP-Bang questionnaire and conducting research based on it.

4.    Furthermore, regarding the CVD risk after 10 years, the authors did not examine the actual progress of 10 years, but only estimated it using the UKPDS Risk Engine.

5.    From the above, the conclusions drawn from these are considered poor of scientific evidence due to problems in both patient setting and evaluation methods.

6.    The AASM strongly recommends polysomnography in patients with CVD, so the authors should do polysomnography properly if considering CVD risk.

Author Response

Point 1: It says "the aim of our study was to assess cardiovascular risk in patients with T2DM and OSA, assessed by the STOP-Bang questionnaire", however, the study may be assessing the cardiovascular risk of patients with T2DM over the age of 18 who are suspected of having OSA by STOP-Bang questionnaire.

Response 1: Thank you for the comment. We rephrased according to the reviewer’s suggestion.

Point 2: The legend in Figure 1 (a) was not found. There is no description of "significant difference" in Figures 1, 4 and 5. Please correct it so that we can understand it only with the figure without reading the text.

Response 2: Thank you for the observation. The legend was generated automatically by the statistical analysis program, but during editing the document the images were overlapped. We made the necessary correction. We also filled in the required information regarding statistically significant differences.

Point 3: Where can I find the contents of Lines 269-273 in Table1?

Response 3: We thank the reviewer for this observation. There was an editing error, the contents of the lines 269-273 are found in Table 3. We made the corrections in the revisited manuscript.

 Point 4: The STOP-Bang questionnaire is useful for OSA screening. However, based on "Anesthesiology. 2008 May; 108 (5): 812-21. Doi: 10.1097 / ALN.0b013e31816d83e4." by Chung F et al., the sensitivity of STOP-Bang is 100% for AHI ≥30, but specificity is 37.0%. This means that there are many false positives, suggesting that many patients have STOP-Bang 3 or higher, even if they are not AHI ≥30.

Response 4: Thank you for the comment. We have added this reference and further discussed this issue in the Discussions section of the manuscript.

Point 5: As a chronic complication of the target patients, there is already a significant difference in ASCVD between the group with STOP-Bang ≥5 and the group with STOP-Bang <5. Therefore, it is not surprising that the risk of predicted CHD after 10 years is higher in the group with STOP-Bang ≥5 than in the group with STOP-Bang <5.

 Response 5: Thank you for pointing this out. In order to avoid confusion and to make the results easier to interpret, we performed a logistic regression adjusting the date for age, gender, smoking status, BMI, WC, NC, T2DM duration, HbA1c and established ASCVD.

Point 6:  In patients with suspected sleep apnea syndrome, the questionnaire alone has limited diagnostic performance, such as the majority of patients without obstructive sleep apnea being misdiagnosed as obstructive sleep apnea.

For this reason, AASM is highly recommended that do not use questionnaires without polysomnography or HSAT, and do polysomnography or HSAT for adults suspected of having moderate or severe obstructive sleep apnea due to symptoms, etc.

Therefore, even if the cost can be saved, there is a problem in diagnosing OSA only with the STOP-Bang questionnaire and conducting research based on it.

Response 6: Thank you for the comment. We have further discussed this issue in the Discussions section of the manuscript.

 Point 7:  Furthermore, regarding the CVD risk after 10 years, the authors did not examine the actual progress of 10 years, but only estimated it using the UKPDS Risk Engine.

Response 7: Thank you for the comment. This study being a cross-sectional one, we could not, at the moment, evaluate the prospective value of STOP-Bang questionnaire regarding cardiovascular events, which we intend to study further in the future. We used the UKPDS Risk Engine as this tool is validated for patients with T2DM. We consider important the fact that STOP-Bang score above 5 is a predictor of cardiovascular risk, as it is easier to perform this questionnaire even in patients with T2DM, the UKPDS Risk engine being more complex and requiring more clinical and paraclinical data that are not always available in primary care, where we recommend the screening using the STOP-Bang questionnaire.

 Point 8:  From the above, the conclusions drawn from these are considered poor of scientific evidence due to problems in both patient setting and evaluation methods.

The AASM strongly recommends polysomnography in patients with CVD, so the authors should do polysomnography properly if considering CVD risk.

Response 8: Thank you for the observation. The patients with STOP-Bang score above 5 were referred to polysomnography, unfortunately only a small proportion of the accepted this investigation. We have modified the aim of the study according to the reviewer’s suggestion, explaining that we assessed the association between OSA risk and cardiovascular risk.

Round 2

Reviewer 2 Report

Thank you for responding to my comments and requests.

I am grateful that many of the concerns have been addressed with additional data and points of discussion.

I still have four concerns:

First, with this revision, the entire manuscript has become very long, 17 pages. For this reason, would you please keep it simple and compact so that it is easy to read?

Second, there are a lot of misspellings or incomplete corrections. There are places where one sentence is redundant. Please proofread in English and correct it.

Table 3. Nonfatal CHR→CHD

 Third, in Figure 1 added in this revision, is the "OSA risk group" the patient group with "STOP-Bang score <5"? If so, it seems that you need to define it when grouping. Furthermore, isn't the number of "OSA risk group" 217?

 Fourth, the following paper is known regarding HOMA-IR of OSA in oriental patients.

Otake K, et al. Glucose Intolerance in Japanese Patients with Obstructive Sleep Apnea. Intern Med. 2009;48(21):1863-8.

Author Response

Point 1: Thank you for responding to my comments and requests.

I am grateful that many of the concerns have been addressed with additional data and points of discussion.

I still have four concerns:

First, with this revision, the entire manuscript has become very long, 17 pages. For this reason, would you please keep it simple and compact so that it is easy to read?

 Response 1: We thank the reviewer for the thoughtful comments. We tried to address the issues that still concern the reviewer. In this regard, we revisited the manuscript, shortening the text where possible as well as combining tables, where possible.                                                                                                     

Point 2: Second, there are a lot of misspellings or incomplete corrections. There are places where one sentence is redundant. Please proofread in English and correct it.

Table 3. Nonfatal CHR→CHD

Response 2: We thank the reviewer for the comment. We revisited the spelling and grammar throughout the manuscript and we tried to address redundancy.

 Point 3: Third, in Figure 1 added in this revision, is the "OSA risk group" the patient group with "STOP-Bang score <5"? If so, it seems that you need to define it when grouping. Furthermore, isn't the number of "OSA risk group" 217?

Response 3: Thank you for pointing out this editing error. We made the necessary corrections in Figure 1. 217 patients had a STOP-Bang score <5 (non-OSA risk group) and 312 had STOP-Bang score ≥5 (OSA risk group).

Point 4: Fourth, the following paper is known regarding HOMA-IR of OSA in oriental patients.

Otake K, et al. Glucose Intolerance in Japanese Patients with Obstructive Sleep Apnea. Intern Med. 2009;48(21):1863-8.

Response 4: Thank you for the suggestion. We discussed this paper in the Discussions section of the manuscript and included it in the reference list.